# Chemometric Differentiation of Sole and Plaice Fish Fillets Using Three Near-Infrared Instruments

**DOI:** 10.3390/foods11111643

**Published:** 2022-06-02

**Authors:** Nicola Cavallini, Francesco Pennisi, Alessandro Giraudo, Marzia Pezzolato, Giovanna Esposito, Gentian Gavoci, Luca Magnani, Alberto Pianezzola, Francesco Geobaldo, Francesco Savorani, Elena Bozzetta

**Affiliations:** 1Department of Applied Science and Technology, Politecnico di Torino, Corso Duca degli Abruzzi 24, 10129 Turin, Italy; alessandro.giraudo@polito.it (A.G.); gentian.gavoci@polito.it (G.G.); francesco.geobaldo@polito.it (F.G.); francesco.savorani@polito.it (F.S.); 2Istituto Zooprofilattico Sperimentale del Piemonte, Liguria e Valle d’Aosta, Via Bologna 148, 10154 Turin, Italy; francesco.pennisi@izsto.it (F.P.); marzia.pezzolato@izsto.it (M.P.); giovanna.esposito@izsto.it (G.E.); elena.bozzetta@izsto.it (E.B.); 3Esselunga S.p.A., Via Giambologna 1, 20096 Limito di Pioltello (MI), Italy; luca.magnani@esselunga.it (L.M.); alberto.pianezzola@esselunga.it (A.P.)

**Keywords:** food fraud, NIR, chemometrics, Guinean sole, European plaice, fish fillets

## Abstract

Fish species substitution is one of the most common forms of fraud all over the world, as fish identification can be very challenging for both consumers and experienced inspectors in the case of fish sold as fillets. The difficulties in distinguishing among different species may generate a “grey area” in which mislabelling can occur. Thus, the development of fast and reliable tools able to detect such frauds in the field is of crucial importance. In this study, we focused on the distinction between two flatfish species largely available on the market, namely the Guinean sole (*Synaptura cadenati*) and European plaice (*Pleuronectes platessa*), which are very similar looking. Fifty fillets of each species were analysed using three near-infrared (NIR) instruments: the handheld SCiO (Consumer Physics), the portable MicroNIR (VIAVI), and the benchtop MPA (Bruker). PLS-DA classification models were built using the spectral datasets, and all three instruments provided very good results, showing high accuracy: 94.1% for the SCiO and MicroNIR portable instruments, and 90.1% for the MPA benchtop spectrometer. The good classification results of the approach combining NIR spectroscopy, and simple chemometric classification methods suggest great applicability directly in the context of real-world marketplaces, as well as in official control plans.

## 1. Introduction

Food mislabelling often results in actual fraud, with potentially heavy and wide impacts on the consumers’ safety and health, as it can potentially affect any type of food commodity that is sold on the market. Moreover, such a practice is, under any circumstances, a violation of the law. Along the fishery production chain, one of the most vulnerable food chains [1], species replacement and mislabelling can often occur [2,3,4,5]. This very specific type of food fraud takes place whenever seafood is wrongly or intentionally labelled with false information about species identity, country of origin, production method, or potential eco-labels [6]. Furthermore, species identification becomes even more challenging when fish is sold in fillets because they often look similar in appearance, taste, and texture. This criminal behaviour misleads consumers, infringes current legislation, and can also result in health issues (e.g., when poisonous or tropical species are consciously sold with false descriptions). Specifically, labelling and traceability of seafood are regulated by the European Parliament resolution of 12th May 2016, on traceability of fishery and aquaculture products in restaurants and retail [7], as well as by the European Parliament Legislative Resolution No 1169/2011 [6], which is focused on providing the final consumer with true and clear information about food products, by clear providing labelling.

To this extent, in order to avoid fraud and to set a trustworthy method for official controls too, nowadays many laboratories rely on genomics techniques, such as DNA barcoding and nucleotide sequencing [2,4,8] whereas traditional methods are based on protein analysis methods such as electrophoretic, chromatographic, or immunological techniques [9]. All these methods, although reliable, are rather time-consuming and expensive and therefore, in practice, limited to a small subset of statistically representative specimens.

From this perspective, quick and reliable methods, able to detect such frauds, are therefore needed for both scientists and decision-makers. Among the most promising analytical techniques in the field of food fraud detection, the application of near-infrared (NIR) spectroscopy has been widely investigated. This is a very well-established analytical technique and has been successfully applied in a large variety of fields (food [10,11,12], feed [13,14,15,16,17] pharmaceutical [18,19], process monitoring [20,21]), but more specifically, it has been used for the detection of food frauds concerning fish products such as actual fish [22,23,24,25,26,27,28,29,30,31], cephalopods [12,32,33], crustaceans [34,35] and mussels [36].

In this study, three NIR instruments were selected to cover a wide and comprehensive range of performances and functionalities, as well as to compare the features of the different devices, with the aim of distinguishing fillets from two flatfish species (Figure 1), namely Guinean sole (*Synaptura cadenati*) and European plaice (*Pleuronectes platessa*). Two devices with different portability and, consequently, applicability in real-world situations for in situ measurements were therefore used, together with a benchtop instrument: a handheld smartphone-controlled device (SCiO by Consumer Physics, Tel Aviv, Israel), a portable laptop-controlled instrument (MicroNIR by VIAVI), and a benchtop spectrometer (MPA by Bruker). Then, one classification model based on the spectral data obtained from each instrument was built, using the partial-least squares–discriminant analysis (PLS–DA, [37]) classification method, one of the most used chemometric tools for classification purposes.

The aim of this study is therefore to propose a new method to distinguish between similar fish species in a fast and reliable way, keeping both sample preparation and data modelling as simple as possible [38]. Moreover, a detailed comparison of the performances of the three NIR instruments is provided, based on the interpretation of actual chemical signals, as recorded by NIR spectroscopy.

## 2. Materials and Methods

This study encompasses the use of three different NIR instruments and simple chemometric tools for exploring the data and building the classification models. The study design is very similar to a previous publication by the authors [12] about cephalopods because they were developed in parallel and using the same instruments.

### 2.1. NIR Instruments

Three NIR instruments with different spectral ranges and resolutions were considered in this study: two portable instruments, namely the handheld SCiO Sensor (version 1.2, Consumer Physics Inc., Tel Aviv, Israel) and the MicroNIR Pro 1700 ES (VIAVI Solutions, San Jose, CA, USA), and one benchtop Fourier transform-NIR spectrometer (Multi-Purpose Analyzer, MPA by Bruker Optics, Ettlingen, Germany) equipped with an optical fibre reflectance probe. The instruments’ technical features are reported in Table 1.

The handheld SCiO sensor was operated through the SCiO smartphone app (The Lab, version 1.3.1.81, by Consumer Physics), which allows controlling the instrument via Bluetooth connection. SCiO’s data management architecture is cloud-based, so each spectra acquisition is first sent from the sensor to the smartphone, and then uploaded and stored on Consumer Physics’ cloud database (Consumer Physics Inc., Tel Aviv, Israel). The experimental acquisition parameters of the SCiO sensor are as follows and cannot be modified by the user: 740–1070 nm (13514–9346 cm^−1^) as spectral range, 10 cm^−1^ resolution, with a typical scan time of less than 5 s.

The portable MicroNIR instrument was controlled via the MicroNIR Pro software (version 2.0, by VIAVI Solutions), operated on a dedicated laptop computer. The experimental acquisition parameters were set as follows: 908–1676 nm (11013–5966 cm^−1^) as spectral range, 12 cm^−1^ resolution, 12.5 µs integration time, 80 Hz scanner velocity, and 200 scans for both sample and background acquisition.

The benchtop MPA instrument was operated via the proprietary software OPUS (version 6.5, by Bruker Optics) using the following setup: 800–2500 nm (12,500–4000 cm^−1^) as spectral range, 12 cm^−1^ resolution, 10 kHz scanner velocity, and 64 scans for both sample and background acquisition. Background scans were performed using the internal reference standard of the instrument.

All NIR instruments were used in diffuse reflectance mode. Instrumental calibration was performed before acquiring the first spectrum and then repeated roughly every ten samples. A visual overview of the raw spectra is given in Figure 2a–c.

### 2.2. Specimens and Spectra Acquisition

Fifty fresh specimens of *Synaptura cadenati* (Guinean sole, hereafter referred to as “SO”) and fifty of *Pleuronectes platessa* (European plaice, hereafter referred to as “PL”) were collected directly from the processing factory of a local large-scale retailer. All specimens had been regularly fished and delivered to the processing plant, where they arrived already in fillets. The whole sampling belonged to six distinct batches collected between October 2019 and February 2020. No physical pre-treatment was performed before spectra acquisition.

The analyses using the two portable devices (SCiO and MicroNIR) were performed directly at the factory. Spectra acquisition using the benchtop MPA instrument was performed after transporting the specimens to the spectroscopic laboratory of the Polytechnic of Turin, maintaining the cold chain (2 ± 2 °C). A total of 3 scans (i.e., “replicates”) were collected for each fish fillet from its fleshy part. No scans from the skin were therefore acquired. All the collected spectra were managed, assessed, pre-treated, pre-processed, and modelled as described in Section 2.3.

### 2.3. Data Processing and Chemometric Modelling

In this section, all relevant information about the data analysis methods used in the study will be detailed, together with the specifications of the software used by the authors.

#### 2.3.1. Raw Data Pre-Treatment and Assessment

The raw data were imported into the MATLAB environment to be assessed, pre-processed, and finally modelled (Section 2.3.2 and Section 2.3.3). To detect and correct potential faults and data integrity problems, an initial quality assessment step was performed. At this stage, the detection of obvious outliers is an important task and the decision to eventually remove them must be substantiated by assessing first if the problematic sample shows any clear defect in its spectral profile. Then, the sample can be also compared to the other two replicates acquired from the same specimen, to determine if any significant difference occurs among them. A description of the replicates analysis procedure can be found in reference [12] by Pennisi et al.

Once all problematic samples were identified, inspected, and finally removed, a new dataset containing only averaged spectra (i.e., obtained from the average taken over each group of replicates) was generated. In this new dataset, each sample corresponds to one specimen or, in other words, one spectrum for each fish fillet was obtained. This pre-treatment procedure was applied to all experimental datasets obtained from the three NIR instruments.

#### 2.3.2. Data Pre-Processing

Prior to both exploratory and classification analyses, all datasets were pre-processed using standard normal variate (SNV, [39]) to remove scattering effects, followed by mean centring. The MPA dataset was also reduced in size from 800–2500 nm (12,500–4000 cm^–1^) to 967–1906 nm (10,350–4000 cm^–1^) to remove the noisy and uninformative far ends of the spectra. A visual overview of the pre-processed spectra is given in Figure 2d–i.

#### 2.3.3. Exploratory Data Analysis

Principal Component Analysis (PCA, [40,41]) was used both to perform the replicates analysis during the data quality assessment (as described in Section 2.3.1) and to explore the pre-processed data before performing the classification modelling.

#### 2.3.4. Classification Modelling

Partial Least Squares-Discriminant Analysis (PLS-DA, [37]) was used to build the classification models to discriminate between sole (SO) and plaice (PL) samples. Test set selection was performed using the Duplex algorithm [42], applied to each class individually to ensure balanced sampling of both classes (i.e., fish species).

Since the classification problem under examination only involves two classes and a discriminant method is applied, the classification results can be inspected from the point of view of one of the two classes. In this case, the results will be reported and interpreted from the point of view of predicting the SO class.

Four parameters were chosen to evaluate the modelling performances [37]:Specificity (Spec): it is the ability to avoid false positives i.e., PL samples wrongly classified as SO.Sensitivity (Sens): it is the ability to avoid false negatives i.e., SO samples wrongly classified as PL.Non-error rate (NER): it is computed as the mean of the sensitivities (one for each class) and corresponds to the overall capability of the model to correctly classify the samples (i.e., both SO and PL).Accuracy (Acc): it is an estimation of the model error, and it is computed as the sum of the true positives (TP, correctly classified sole samples) and true negatives (TN, correctly classified plaice samples) divided by the total number of samples.

As a consequence of having a two-classes model, Spec and Sens values of the PL class are going to be the “mirrored” version of the values corresponding to the SO class: in other words, the specificity of the SO class is equal to the sensitivity of the PL class, and vice versa.

The most important variables of each classification model were inspected and identified using the variable importance in projection (VIP) scores [43].

### 2.4. Software and Toolboxes

The whole data analysis workflow was developed under the MATLAB environment (versions 2017 and 2021, The Mathworks, MA, USA). PCA exploratory analysis and PLS-DA classification modelling were performed using the functions of the PLS_Toolbox (versions 8.6 and 8.9, Eigenvector Research Inc., Manson, WA, USA) software package. Calibration and test set selection was performed using the Duplex algorithm by Michał Daszykowski (https://www.researchgate.net/publication/305536996, MATLAB code for the Duplex uniform subset selection algorithm, last accessed 25 March 2022). In-house written routines were used to import, manage, and assess the raw data and organize the data analysis workflow. The smartphone app “The Lab” (version 1.2.1 for most of the experimental data acquisition; https://play.google.com/store/apps/details?id=com.consumerphysics.researcher or https://apps.apple.com/us/app/the-lab-dev-toolkit-for-scio/id965758603, last accessed 19 April 2022) used to control the SCiO sensor was installed on an Android phone. SCiO spectra were managed and downloaded from the web interface of “The Lab”.

## 3. Results

The results are structured as follows: in Section 3.1 an overview of the information obtained from each dataset by means of PCA is given and discussed; in Section 3.2 the PLS–DA classification results and performances of each dataset are reported and briefly discussed. Finally, in Section 4 (Discussion), the classification results are interpreted and discussed from the point of view of the spectral variables that resulted most important in the classification modelling step.

### 3.1. Exploratory Analysis Results

For the sake of clarity, only the best separation tendencies obtained with PCA are reported (Figure 3). A good separation tendency was found with the SCiO dataset, as shown by PC1 and PC2 combined (Figure 3a). Additionally, the combination of PC2 and PC4 (not shown) provided information related to the SO-PL separation, and the good classification modelling performances confirmed the presence of such useful information (Table 2). The separation tendency provided by the MicroNIR dataset (Figure 3b) results is less clear, especially if compared to the other two spectroscopic techniques (Figure 3a,c). Moreover, in this case, PC3 turned out to be the component most related to the SO-PL separation, with only 7.24% of explained variance. Regarding the MPA dataset, a separation tendency could be detected along PC2 (30.94% of explained variance, Figure 3c). It is important to state that two outliers were identified and removed, after assessing their spectral look (reported in the Appendix A): for these two samples, serious spectral defects were found, and therefore it was decided to remove them.

### 3.2. Classification Analysis Results: Prediction of Sole (SO) vs. Plaice (PL) Samples

The PLS-DA classification performance results of the model are reported in Table 2. All classification performances resulted very high, with minor issues regarding the sensitivities of the Micro NIR and MPA datasets. The classification model of the SCiO dataset appears to be the best and most consistent one, since it scored very high (indeed, almost perfectly) in all the computed prediction measures, and concerning the test set all its values are above 90%. Moreover, the SCiO model is the simplest one, as it was fitted using 7 latent variables, as opposed to the other two models which required 11 latent variables. The accuracy in prediction, however, appears to be the same for both the SCiO and MicroNIR instruments, while the MPA spectrometer yielded lower performances (90.1% accuracy).

The absolute numbers of misclassified samples in prediction (from the test set) turned out to be:SCiO dataset: 2 out of 34 (one PL predicted as SO, one SO predicted as PL).MicroNIR dataset: 2 out of 34 (2 PL predicted as SO).MPA dataset: 3 out of 33 (3 SO predicted as PL).

The most influential wavelengths, represented in Figure 4a–c using the VIP scores and further discussed in Section 4, appear to be:

SCiO dataset: mainly three areas, i.e., 740–780 nm, 875–920 nm, 1045–1070 nm.MicroNIR dataset: several individual peaks and bands, mostly along the wavelength range 1000–1400 nm.MPA dataset: several individual peaks in the intervals 1010–1060 nm, 1290–1350 nm, and 1700–1910 nm; a wide “missing” signal can be identified in the central region of the spectrum, within the range 1380–1615 nm, but this region was removed due to its high content of noisy signals.

## 4. Discussion

### Classification Analysis Results: VIP Scores Interpretation

From the VIP scores plots reported in Figure 4, it is possible to identify the most influential spectral variables in relation to the classification problem, i.e., those wavelengths that contributed the most to the classification model for the differentiation between the two classes. Several signals across the acquired spectral ranges were identified and assigned to classes of compounds (Table 3).

As reported by Liu et al. [24], water is the main constituent of fish flesh, generally accounting for about 66–81% of the weight of a fresh fish fillet. For all three NIR instruments, the VIP plots highlighted signals related to water: the 3rd overtone at 760 nm (SCiO) and the 2nd overtone at 1450 nm (SCiO, MicroNIR, MPA) [22,24,44]. It is important to note that the water signal at 1450 nm recorded by MPA suffered saturation issues, and therefore the range 1380–1615 nm was removed from the dataset: however, the intense and wide signal related to water is still partially present, and it resulted relevant in the classification model.

Several signals related to aliphatic groups were also identified: peaks at 1120 nm and 1150 nm (MicroNIR), two peaks at 1210 nm [22,44] and 1360–1370 nm related to the 2nd overtone of C–H stretching (MicroNIR), and multiple absorptions in the range 1700–1910 nm linked to the first overtone of C–H stretching [44] of fats [24] (MPA). These signals can arise from different compounds (such as fats) and biological structures which constitute the fish flesh.

In addition to fats and aliphatic compounds in general, signals arising from functional groups containing nitrogen and therefore related to proteins were also identified: in the range 875–920 nm (SCiO), at 1050 nm the 3rd overtone of RNH_2_ (SCiO partially cut, MicroNIR, MPA). More specifically, the shoulder signal at about 1600 nm in the MPA VIP scores (Figure 4c) can be linked to protein fraction absorption (1st and 2nd overtones of N–H, and the combination of N–H and C=O signals [24,44]).

Two unassigned groups of signals also resulted relevant for the classification problem: a peak at 1310 nm for the MicroNIR and an indented band at 1290–1350 nm for the MPA.

Finally, since the SCiO sensor also covers a small part of the visible range, the intense absorption band that can be seen at <750 nm in Figure 4a can be interpreted as absorption of the colour red, which might be related to visual features of the flesh (its shades of red) due to the presence of compounds related to blood. A slight colour difference could already be spotted by the naked eye (see Figure 1) but assessing such a small feature can prove to be unreliable as well as subjective.

Despite the rather big differences among the three instruments, the recovered information appears coherent across all of them. Similar classification performances were also obtained for all datasets, thus suggesting that the discussed variables represent a robust subset of meaningful signals.

## 5. Conclusions

Selling mislabelled fish products is one of the most common forms of fraud, misleading consumers, infringing current legislation, and potentially resulting in health issues. In this field, reference, and gold standard analyses to identify animal species generally require rather long processing times, not to mention that quick decision-making is fundamental in preventing criminal actions.

For this reason, this study investigated the performances of three NIR spectroscopy devices in distinguishing between two very similar flatfish species, which is to all intents and purposes, an authentication issue. Additionally, the goal of setting up a reliable set of models to be used in quality and official controls was also pursued.

NIR spectroscopy surely represents a quick and cost-effective technique that offers many advantages compared to traditional analytical methods, and its application in this study is in accordance with addressing overcoming both the time and cost limitations of the traditional authentication procedures. NIR spectroscopy needs minimal or no sample preparation, it is non-destructive, but most importantly it can be used directly in real-world situations, making it the ideal tool for performing quick and massive screening analyses.

All investigated NIR devices proved to be able to acquire relevant information to build classification models with good accuracy, therefore proving that NIR spectroscopy can be considered a powerful technology in the fight against food fraud. Interestingly, the most consistent performances were obtained using the handheld and portable devices (the SCiO sensor above all), suggesting that simple and practical analyses could be deployed for screening directly on the production lines. Based on these findings, food fraud could be detected using portable NIR devices directly in the marketplace by the consumers at the moment of purchase, as well as by the reselling companies when dealing with suppliers. On the other hand, official control laboratories would still serve their legal purpose of confirming suspect fish origins, by operating in close connection with larger screening operations, using traditional techniques.

## Figures and Tables

**Figure 1 foods-11-01643-f001:**
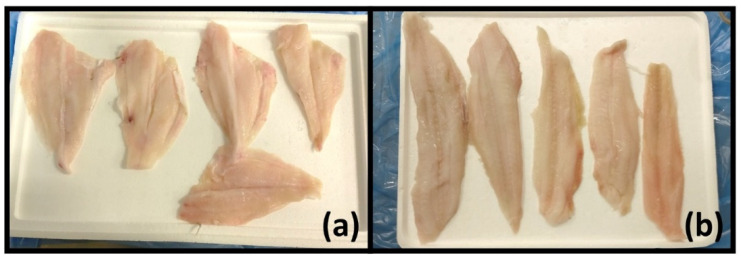
Fillets of (**a**) Guinean sole (*Synaptura cadenati*) and (**b**) European plaice (*Pleuronectes platessa*).

**Figure 2 foods-11-01643-f002:**
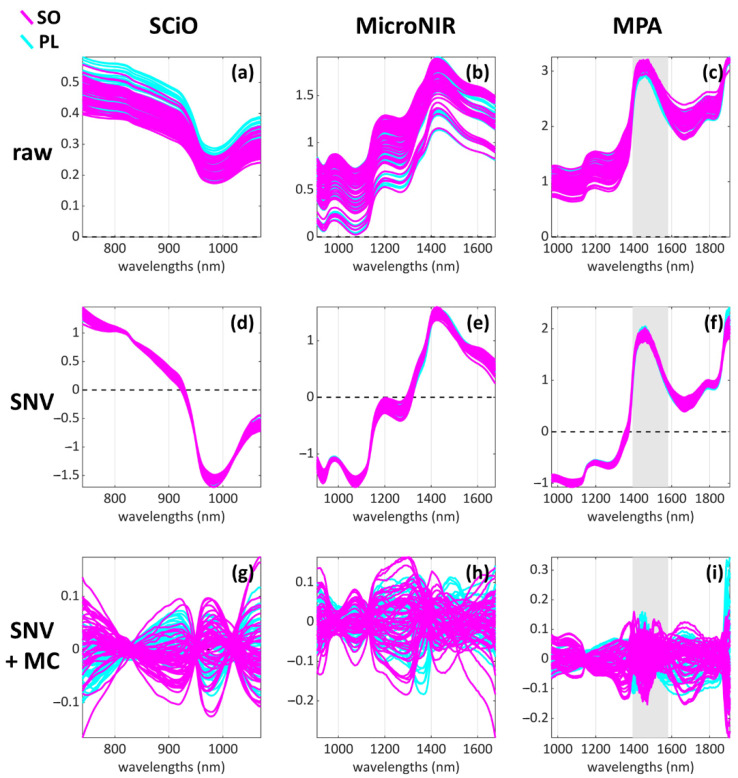
Visual representation of the datasets of the study: (**a**–**c**) the raw data, (**d**–**f**) the SNV-pre-processed data and (**g**–**i**) the SNV and mean centred (MC) data.

**Figure 3 foods-11-01643-f003:**
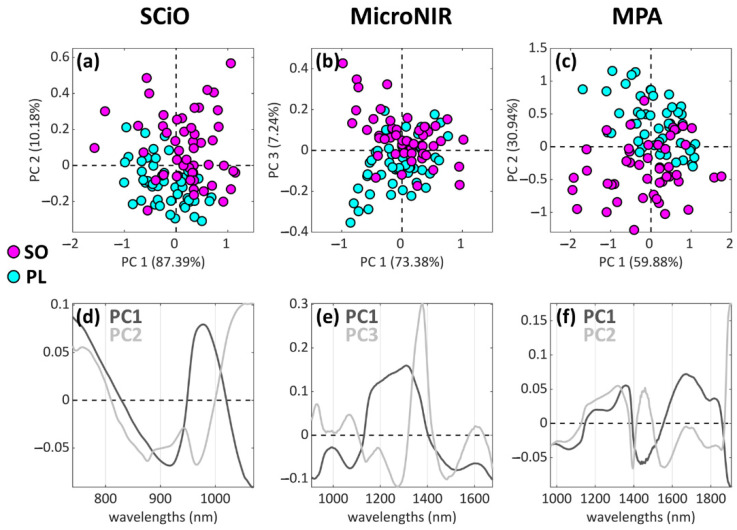
Most relevant PCA scores plots (**a**–**c**) in terms of SO-PL class separation, and the corresponding loadings plots (**d**–**f**). The three columns refer to, respectively, the (**a**) SCiO, (**b**) MicroNIR, and (**c**) MPA datasets.

**Figure 4 foods-11-01643-f004:**
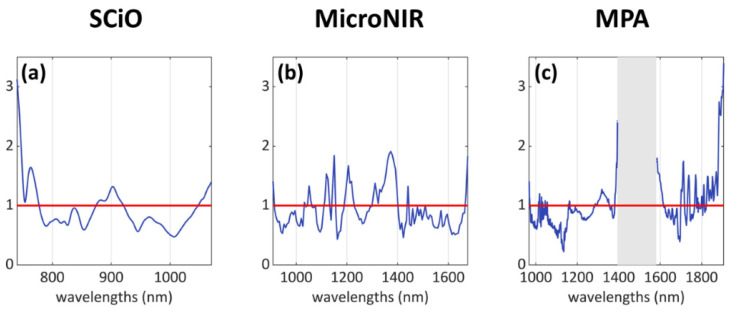
Variable importance in projection (VIP) scores of the three PLS-DA classification models: (**a**) SCiO results, (**b**) MicroNIR results, and (**c**) MPA results. The traditional VIP threshold = 1 is represented in red.

**Table 1 foods-11-01643-t001:** Main technical features of the NIR spectrometers (also reported in [12]).

	Manufacturer	Size (cm, W × L × H)	Weight (g)	Cost ($)	Spectral Range (nm)	Spectral Range (cm^–1^)
SCiO	Consumer Physics	1.5 × 4 × 6.5	<50	<5000	740–1070	13,514–9346
MicroNIR	VIAVI	4.6 × 4.6 × 5	64	≈35,000	908–1676	11,013–5966
MPA	Bruker Optics	37.5 × 59.3 × 26.2	3500	≈150,000	800–2500	12,500–4000

**Table 2 foods-11-01643-t002:** Classification results: sole (SO) vs. plaice (PL), point of view for interpreting the values is the prediction of the SO class. All values are in percentage and those in bold score above 90%.

	SCiO					MicroNIR					MPA				
	LVs	Spec	Sens	NER	Acc	LVs	Spec	Sens	NER	Acc	LVs	Spec	Sens	NER	Acc
Cal		**100**	**100**	**100**	**100**		**100**	**100**	**100**	**100**		**100**	**100**	**100**	**100**
CV	7	**100**	**97.1**	**98.5**	**98.5**	11	**94.1**	**96.9**	**95.5**	**95.4**	11	**91.2**	**93.5**	**92.4**	**92.3**
Test		**94.1**	**94.1**	**94.1**	**94.1**		**100**	89.5	**94.7**	**94.1**		**100**	84.2	**92.1**	**90.9**

**Table 3 foods-11-01643-t003:** Signal assignments of the most influential variables identified using the VIP scores of the PLS–DA models.

	SCiO	MicroNIR	MPA
Water	760 nm, 3rd overtone 1450 nm, 2nd overtone [22,24,44]	1440 nm, 2nd overtone [22,24,44]	1380–1615 nm, 2nd overtone
Proteins	875–920 nm 1045–1070 nm	1050 nm, RNH_2_ 3rd overtone	1010–1060 nm, RNH_2_ 3rd overtone 1600 nm, N–H 1st and 2nd overtones + N–H and C=O combination band [24,44]
Aliphatic	/	1120 nm, 1150 nm 1210 nm C–H stretching 2nd overtone [22,44] 1360 nm C–H stretching 2nd overtone	1700–1910 nm 1st overtone C–H stretching [44] of fats [24]
Other	<750 nm, related to red adsorption	1310 nm-unassigned	1290–1350 nm-unassigned

## Data Availability

Data available on request.

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
