# Peer review of "Chemometric Differentiation of Sole and Plaice Fish Fillets Using Three Near-Infrared Instruments"

_foods, 2022, doi:10.3390/foods11111643_

Round 1

Reviewer 1 Report

The authors presented a well-written research paper dealing with multivariate discriminant analysis coupled with near-infrared spectroscopy to distinguish two flatfish species. Analytical approaches combining chemometric tools with spectroscopy data have been playing an important role in falsification detection, especially in the food industry. The authors showed the spectral bands that most contributed to the classification models and demonstrated the feasibility of the proposed method even using data from different spectrometers. Therefore, the general topic of this paper is very relevant and has a valuable contribution to food authentication.

The authors comprehensively introduced the research subject, described the applied methodology in enough detail to be fully understood, showed their research results effectively, and discussed them appropriately. I really enjoyed the reading, and I am pleased to recommend addressing minor suggestions:

  • I didn’t understand the paragraph from line 86 to line 90. It likes out of context.
  • I suggest a figure with NIR spectra. This helps to see the data nature (the existence of specific bands for each fish species as well as the difference among the spectrometers).
  • In figure 2, it is observed separation tendencies with PCA. I would recommend a figure with loadings to verify what variables contribute to these separations.

Author Response

Chemometric differentiation of sole and plaice fish fillets using three near-infrared instruments [foods-1717954]

Nicola Cavallini*, Francesco Pennisi, Alessandro Giraudo, Marzia Pezzolato, Giovanna Esposito, Gentian Gavoci, Luca Magnani, Alberto Pianezzola, Francesco Geobaldo, Francesco Savorani, Elena Bozzetta

Response to Reviewer #1

The authors presented a well-written research paper dealing with multivariate discriminant analysis coupled with near-infrared spectroscopy to distinguish two flatfish species. Analytical approaches combining chemometric tools with spectroscopy data have been playing an important role in falsification detection, especially in the food industry. The authors showed the spectral bands that most contributed to the classification models and demonstrated the feasibility of the proposed method even using data from different spectrometers. Therefore, the general topic of this paper is very relevant and has a valuable contribution to food authentication.

The authors comprehensively introduced the research subject, described the applied methodology in enough detail to be fully understood, showed their research results effectively, and discussed them appropriately. I really enjoyed the reading, and I am pleased to recommend addressing minor suggestions

We would like to thank the reviewer for the positive feedback and the clearly stated suggestions. We will address them individually:

  1. I didn’t understand the paragraph from line 86 to line 90. It likes out of context.

The paragraph in question is a “leftover” from the Foods journal template, and it was mistakenly left in the final text! It has been removed, thank you for noticing.

  1. I suggest a figure with NIR spectra. This helps to see the data nature (the existence of specific bands for each fish species as well as the difference among the spectrometers).

Thank you for the suggestion, a figure with the raw NIR spectra, and the preprocessed version, coloured according to the two classes (fish species) was added to the manuscript as Figure 2. This also addresses a similar request from Reviewer #2.

  1. In figure 2, it is observed separation tendencies with PCA. I would recommend a figure with loadings to verify what variables contribute to these separations.

This suggestion makes sense also regarding the previous comment, and it will help referring to the original variables for signal interpretation. We therefore added the corresponding loadings to Figure 2 (now Figure 3, with the addition of the new figure), which now has a top row with the PC scores plots and a bottom row with the corresponding loadings.

Reviewer 2 Report

This study  proposed a new method to distinguish fish species in a fast and reliable way and  provided a detailed comparison of the performances of the three NIR instruments. It is useful but needs further modification.

1. In this study, the author established a method based on infrared spectroscopy. What traditionnal methods are commonly used for i distinguishing fish fillets?
2. I suggest that the author should provide the original spectra of the two fish species. If there are obvious differences in the original spectra, can they be distinguished without data processing and modeling? If the original spectral is similar, why does the author choose only one method (SNV) for data processing in this study? why does the author establish only one discriminant model (PLS-DA)? If other data processing methods or models are selected, will the accuracy of the developed in this study be further improved?
3. What kind of fish fillets does the spectral information in Figure 3 represent? Why not show the data of another fish fillet? Are the VIP scores of the two kinds of fillets consistent?

Author Response

Chemometric differentiation of sole and plaice fish fillets using three near-infrared instruments [foods-1717954]

Nicola Cavallini*, Francesco Pennisi, Alessandro Giraudo, Marzia Pezzolato, Giovanna Esposito, Gentian Gavoci, Luca Magnani, Alberto Pianezzola, Francesco Geobaldo, Francesco Savorani, Elena Bozzetta

Response to Reviewer #2

This study proposed a new method to distinguish fish species in a fast and reliable way and provided a detailed comparison of the performances of the three NIR instruments. It is useful but needs further modification.

  1. In this study, the author established a method based on infrared spectroscopy. What traditional methods are commonly used for i distinguishing fish fillets?

Thank you for the comment. As suggested, we have added a brief paragraph in the Introduction, from line 51 to line 57, and three new bibliographic references, to address this lack of information.

  1. I suggest that the author should provide the original spectra of the two fish species. If there are obvious differences in the original spectra, can they be distinguished without data processing and modeling?

The suggestion of providing the original spectra was also proposed by the Reviewer #1, and a new Figure 2 was added to the manuscript: the raw spectra, SNV and SNV + mean centre preprocessed spectra are now depicted in the figure, making it clear that only subtle differences between the two species can be visually noticed. We agree with the Reviewers that such a figure adds clarity to the manuscript, allowing for a better comparison with the PCA loadings (updated version of Figure 2, which is now Figure 3) and the VIP scores of Figure 3 (now Figure 4).

SNV + mean centre preprocessing seems to perform better to this regard, but the rather small highlighted differences support the need for modelling in order to systematically distinguish between the two fish species.

If the original spectral is similar, why does the author choose only one method (SNV) for data processing in this study? Why does the author establish only one discriminant model (PLS-DA)?

Due to the scopes of the study, one of the simpler and more standard preprocessing method (SNV + mean centre) was applied to all datasets, also to allow for easier comparisons. The same reasoning applies to the choice of using PLS-DA as a classification method. Furthermore, the obtained results are already fully addressing the main aims of this research study, making it less relevant to test all the other preprocessing combinations.
In addition, the point of view of investigating and comparing the modelling performances of three different analytical instruments, also drove the decision of “keeping it simple”, together with the need for clearer and more direct communication of the promising results also to a wider audience, which might also include experts from the regulatory authorities.

If other data processing methods or models are selected, will the accuracy of the developed in this study be further improved?

Due to the abovementioned communication scope and to the fact that three datasets were available, one for each analytical instrument tested, the authors decided not to systematically assess an array of preprocessing methods: that would have been a more complex paper to make it clear and concise. However, we think that additional test, encompassing different preprocessing methods, would surely be needed in the case of further developments and implementations of the methodological approach described in the study. Before reaching a real-world application, the models would need fine tuning and testing.

  1. What kind of fish fillets does the spectral information in Figure 3 represent? Why not show the data of another fish fillet? Are the VIP scores of the two kinds of fillets consistent?

Figure 3 (now Figure 4) represents the VIP scores of the PLS-DA models. The VIP method was used for determining which original variables (the wavelengths, in this case) are the most important (relevant) for the prediction capability of the PLS-DA models. There is only one set of VIP scores for each model, so we cannot distinguish or make comparisons among profiles of different fish fillets using such data. Nevertheless, we are ready to provide the Reviewer with any other kind of information that could help making the paper’ message easier to understand.
